# Characterization, Genome Sequencing, and Development of a Rapid PCR Identification Primer for *Fusarium oxysporum* f. sp. *crocus*, a New *forma specialis* Causing Saffron Corm Rot

**DOI:** 10.3390/plants13223166

**Published:** 2024-11-11

**Authors:** Zhenyu Rong, Tingdan Ren, Junji Yue, Wei Zhou, Dong Liang, Chuanqing Zhang

**Affiliations:** 1College of Advanced Agricultural Sciences, Zhejiang Agriculture and Forest University, Hangzhou 311300, China; 2Extension Centre of Agriculture Technology of Jiande, Hangzhou 311600, China; 3College of Pharmacy, Zhejiang Chinese Medical University, Hangzhou 310053, China; zhouwei19810501@163.com

**Keywords:** saffron corm rot, *Fusarium oxysporum*, host specialization, *SIX* gene, rapid identify

## Abstract

Saffron corm rot (SCR), the most serious disease affecting saffron, has been confirmed to be caused by *Fusarium oxysporum* in previous studies. Compared to other fungal species, *F. oxysporum* exhibits host specialization, a special phenomenon associated with the secreted in xylem (*SIX*) genes. This study examined the pathogenicity specialization of *F. oxysporum* isolated from saffron corms with SCR disease. The results showed that this *F. oxysporum* strain was strongly pathogenic to saffron corms, causing SCR; weakly pathogenic to the corms of freesia, which is in the Iridaceae family along with saffron; and not pathogenic to watermelon, melon, and tomato. Other *formae speciales* of *F. oxysporum* were not pathogenic to saffron corms. This suggests that *F. oxysporum* saffron strains exhibit obvious pathogenicity specialization for Iridaceae spp. Subsequently, the *F. oxysporum* saffron strain (XHH35) genome was sequenced, and a comparative genomics study of XHH35 and three other *formae speciales* was conducted using OrthoVenn3. XHH35 contained 90 specific genes absent in the other three *formae speciales*. These genes are involved in certain key biological processes and molecular functions. Based on BLAST homology searching, the *F. oxysporum* saffron strain (XHH35) genome was predicted to contain seven *SIX* genes (*SIX 4*, *SIX 6*, *SIX 7*, *SIX 10*, *SIX 11*, *SIX 12*, and *SIX 14*) highly homologous to *F. oxysporum* f. sp. *lycopersici*, which was verified using polymerase chain reaction (PCR) amplification. The corresponding individual phylogenetic tree indicated that the *F. oxysporum* saffron strain (XHH35) showed a separate branch with different *formae speciales*. This study is the first-ever report of *F. oxysporum* f. sp. *crocus*, a new *forma specialis*. Based on the specificity of its *SIX* genes, the *SIX 10* gene was selected to further establish a rapid identification technique for *F. oxysporum* f. sp. *crocus*, which will be useful in future research.

## 1. Introduction

Saffron (*Crocus sativus* L.) is a medicinal plant [1]. In China, saffron is recognized as a rare Chinese herbal medicine that improves blood circulation, cools the blood, detoxifies, relieves depression, improves immunity, and prevents cardiovascular disease [2]. Hangzhou, specifically Sandu Town in Jiande City (29°54′ N, 119°57′ E), is considered the birthplace of saffron and is the main saffron production area in China. Sandu Town is known as the “hometown of saffron in China” and accounts for 80% of the national saffron yield [3].

Saffron corm rot (SCR) has become the most common and serious disease in saffron production in recent years. SCR directly reduces the quality and yield of stigma and affects the production of corms [1]. The typical symptoms of SCR in the field include leaf yellowing and wilting on the ground, as well as atrophied rotting corms underground. Although a variety of fungi, including *Fusarium* [4], *Aspergillus* [5], and *Penicillium* [6], have been isolated from saffron corms with SCR, research has confirmed that *Fusarium oxysporum* is the pathogenic fungus causing SCR [4].

*F. oxysporum* is a widely distributed soil-borne pathogenic fungus that causes root and corm rot in hundreds of cultivated plants, resulting in significant economic losses [7]. Compared with other *Fusarium* species, the pathogenicity of *F. oxysporum* varies in different types of crops and different varieties of the same crop, exhibiting a high degree of host specificity [8]. For example, the pathogen forms of *F. oxysporum* have distinct host specialization on watermelon and melon, both of which are members of the cucurbit family [9]. *F. oxysporum* f. sp. *freesia* infects freesia, which belongs to Iridaceae, the iris family [10], as does saffron. However, the causal agent of SCR, *F. oxysporum*, has an unknown host specialization in saffron.

The identification of pathogenic fungi host specialization has traditionally relied on pure culture methods. However, traditional methods are sometimes unable to accurately identify pathogenic fungi due to the influence of factors such as the culture medium and culture conditions [11]. The secreted in xylem (*SIX*) genes have been utilized for the identification of *F. oxysporum* host specializations. The first *SIX* protein reported was a 12-KDa cysteine-rich protein named *SIX 1*, identified by Rep et al. in the xylem of tomato plants [12]. *F. oxysporum* can secrete a number of cysteine-rich *SIX* proteins encoded by *SIX* genes to initiate pathogenicity when it infests the xylem of the host [13]. Although 14 *SIX* genes have been described in the *F. oxysporum* f. sp. *lycopersici* (*Fol*) lineages to date, not all 14 *SIX* genes are carried by other host specializations [14,15]. In fact, as demonstrated by van Dam et al., the host specializations of *F. oxysporum* each harbor a unique combination of effectors that can be employed to distinguish different host-specialized lineages [16].

This study aimed to experimentally determine the existence of a host specialization of *F. oxysporum* in saffron. Moreover, this work developed a rapid identification method for the potential host specialization, providing a theoretical basis for the subsequent identification of host specializations in saffron.

## 2. Results

### 2.1. Pathogenicity Specialization

A pathogenicity test was conducted of the *F. oxysporum* saffron strain XHH35 (strain number) on melon seedlings, cucumber seedlings, tomato seedlings, saffron corms, and freesia corms (Figure 1). The test showed that melon, cucumber, and tomato seedlings did not develop disease in the root system after root dipping treatment, the leaf growth was healthy, and no significant differences were noted compared with the control group, indicating that *F. oxysporum* was not pathogenic to melon, cucumber, or tomato. In contrast, the inoculation sites of both saffron and freesia corms inoculated with the spore solution exhibited lesions; *F. oxysporum* was strongly pathogenic to saffron corms, in which it caused SCR and weakly pathogenic to freesia corms.

The pathogenicity test of the other host-specialized strains of *F. oxysporum* (*F. oxysporum* f. sp. *lycopersici*, *F. oxysporum* f. sp. *niveum*, and *F. oxysporum* f. sp. *melonis*) on saffron corms (Figure 2). The results showed that none of the saffron corms inoculated with the spore solutions of the other host-specialized strains showed lesions at the inoculation site, and there was no significant difference from the non-inoculated group. This indicates that the other host-specialized strains of *F. oxysporum* are not pathogenic to saffron corms.

In conclusion, *F. oxysporum* saffron strains are pathogenicity specialized to *Iridaceae* spp.

### 2.2. Genome Sequencing

The de novo genome assembler suites NextDenovo v2.4.0 and NextPolish v1.3.1 (both developed by NextOmics; https://github.com/Nextomics, accessed on 10 May 2024.) were employed to generate a genome assembly of high accuracy and continuity. First, a draft genome assembly was assembled using NextDenovo based on PacBio long reads; then, it was polished using NextPolish based on both PacBio long reads and Illumina short reads. Finally, a 56.89-Mb genome assembly (GC content 48%) was obtained that consisted of 34 contigs with a contig N50 of 4.24 Mb and a contig N90 of 2.23 Mb (Table 1).

Using OrthoVenn3 (https://orthovenn3.bioinfotoolkits.net, accessed on 16 August 2024.), the genomes of XHH35, *Fusarium oxysporum* f. sp. *lycopersici*, *Fusarium oxysporum* f. sp. *melonis* and *Fusarium oxysporum* f. sp. *cubense* genomes for protein clustering analysis (based on the OrthoMCL algorithm and e-value of 1 × 10^−2^ was used). A comparative study of the protein clustering analysis revealed there are 12,330 highly conserved orthologous clusters in XHH35, *F. oxysporum* f. sp. *lycopersici*, *F. oxysporum* f. sp. *melonis* and *F. oxysporum* f. sp. *cubense*. (Figure 3a). Notably, XHH35 contained 90 specific protein clusters (including 245 genes) that were absent in the other three host-specialized strains. After GO clustering analysis of these 90 specific protein clusters, we found that seven genes each were clustered into the transmembrane transport and transcription (DNA-templated); six genes were clustered into the oxidoreductase activity (acting on paired donors, with incorporation or reduction of molecular oxygen); five genes were clustered into the polyketide biosynthetic process; three genes were clustered into the regulation of transcription (DNA-templated); two genes each were clustered into the sporulation resulting in formation of a cellular spore, carbohydrate transport, positive regulation of transcription from RNA polymerase II promoter and carbohydrate metabolic process. These included genes involved in critical biological processes and affect critical molecular functions (Table 2, Figure 3b).

### 2.3. Amplification of the SIX Gene

Based on BLAST homology searching, the *F. oxysporum* saffron strain (XHH35) genome was predicted to contain seven *SIX* genes (the *SIX 4*, *SIX 6*, *SIX 7*, *SIX 10*, *SIX 11*, *SIX 12*, and *SIX 14* genes) that are highly homologous to *Fol* (Appendix A).

Polymerase chain reaction (PCR) amplification to further confirm the presence of these genes in XHH35. The agarose gel electrophoresis gel plot results for the amplification of the *F. oxysporum* saffron strains of genes *SIX 1* through *SIX 14* genes are shown in Figure 4. It was found that the PCR products of the correct size were amplified from the DNA template of *SIX 4*, *SIX 6*, *SIX 7*, *SIX 10*, *SIX 11*, *SIX 12*, and *SIX 14* genes. However, the PCR products of the correct size were not amplified from the DNA template of *SIX 1*, *SIX 2*, *SIX 3*, *SIX 5*, *SIX 8*, *SIX 9,* and *SIX 13* genes.

### 2.4. SIX Homologous Gene Evolutionary Tree Analysis

The *SIX 4*, *SIX 6*, *SIX 7*, *SIX 10*, *SIX 11*, *SIX 12*, and *SIX 14* gene sequences of the *F. oxysporum* saffron strain (XHH35) that amplified a single target band in PCR were collected separately, and the *SIX* gene sequences of *F. oxysporum* strains from different hosts were also categorized and downloaded according to *SIX 4*, *SIX 6*, *SIX 7*, *SIX 10*, *SIX 11*, *SIX 12*, and *SIX 14* from the National Center for Biotechnology Information (NCBI) (Appendix A). A total of seven gene evolutionary tree images were obtained using the neighbor-joining, maximum likelihood, and maximum parsimony methods for gene evolutionary tree construction after sequence comparison using MEGA 11. The results are shown in Figure 5.

In the gene evolutionary trees of *SIX 4*, *SIX 6*, *SIX 7,* and *SIX 11* genes, XHH35 was on a single branch, suggesting that the *SIX 4*, *SIX 6*, *SIX 7,* and *SIX 11* genes of XHH35 are more distantly related than those of other host-specialization strains (Figure 5a–c,e).

In the gene evolutionary trees of *SIX 10*, *SIX 12,* and *SIX 14* genes, XHH35 was clustered in the same branch as the *F. oxysporum* f. sp. *freesia*, suggesting that the *SIX 10*, *SIX 12,* and *SIX 14* genes of XHH35 with *F. oxysporum* f. sp. *freesia* are closely related (Figure 5d,f,g).

### 2.5. Establishment of Rapid Detection Methods

The *SIX 10* gene of *F. oxysporum* saffron strain was selected for the design of specific primers due to its high specificity compared to other host-specialized strains. Sequence comparison was performed of the amplified *SIX 10* gene of *F. oxysporum* saffron strains with the *SIX 10* genes of other host-specialized strains (Figure 6).

Based on the results of the comparison, eight pairs of specific primers were designed using Primer Premier v5.0 (Table 3). It was found that primer 4 could amplify all of the *F. oxysporum* saffron strains bands of about 275 bp in length but could not amplify the *F. oxysporum* f. sp. *niveum*, *F. oxysporum* f. sp. *fragariae, F. oxysporum* f. sp. *melonis*, and *F. oxysporum* f. sp. *lycopersici*; primers 1 and 3 could amplify some of the *F. oxysporum* saffron strains but could not amplify the *F. oxysporum* f. sp. *niveum*, *F. oxysporum* f. sp. *fragariae*, *F. oxysporum* f. sp. *melonis*, and *F. oxysporum* f. sp. *lycopersici*; and primers 2, 5, 6, 7, and 8 could amplify the *F. oxysporum* f. sp. *niveum*, *F. oxysporum* f. sp. *fragariae*, *F. oxysporum* f. sp. *melonis*, and *F. oxysporum* f. sp. *lycopersici*. This indicates that primer 4 can specifically screen out eight *F. oxysporum* saffron strains (Figure 7, Appendix A).

This study further validated the specificity of primer 4 against other strains (Figure 8), which revealed that this primer could only specifically amplify *F. oxysporum* saffron strains and had no target bands for other strains. In conclusion, this primer can be employed as a specific primer for the rapid identification of *F. oxysporum* saffron strains.

## 3. Discussion

*Fusarium oxysporum* is a non-negligible pathogen that causes plant diseases and can infect a wide range of economically important plants around the world, including saffron [17,18,19], resulting in huge economic losses. A phenomenon has been observed in the pathogenic strains of *F. oxysporum*, namely “host specialization”. The pathogenicity of *F. oxysporum* strains varied among different crop species and among different varieties of the same crop [7,14,20]. The results of the pathogenicity experiments in this study also confirm this conclusion. The *F. oxysporum* strain isolated from saffron corms was strongly pathogenic to saffron corms and weakly pathogenic to the corms of freesia, which belongs to the Iridaceae family along with saffron but was not pathogenic to watermelon, melon, and tomato. Other *formae speciales* of *F. oxysporum* (*F. oxysporum* f. sp. *lycopersici*, *F. oxysporum* f. sp. *niveum*, and *F. oxysporum* f. sp. *melonis*) are not pathogenic to saffron corms. This suggests that *F. oxysporum* saffron strains have obvious pathogenicity specialization for Iridaceae spp.

Subsequently, the genome of *F. oxysporum* saffron strain (XHH35) was sequenced and a comparative genomics study was conducted on XHH35 and three additional host-specialized strains of *F. oxysporum* (*F. oxysporum* f. sp. *lycopersici*, *F. oxysporum* f. sp. *cubense*, and *F. oxysporum* f. sp. *melonis*) using OrthoVenn3. The results showed that there are 12,330 highly conserved orthologous clusters in XHH35 and three additional host-specialized strains of *F. oxysporum*. In particular, there were 90 orthologous groups identified that were specific in XHH35, which contained 245 genes. These genes involved the function of oxidoreductase activity (acting on paired donors, with incorporation or reduction of molecular oxygen) and involved the transmembrane transport, transcription (DNA-templated), sporulation resulting in the formation of a cellular spore, and so on biological process.

Because the *SIX* homologous genes in different host-specialized *F. oxysporum* strains are not identical, studies have further confirmed that there is a link between *F. oxysporum* host specificity and *SIX* genes [21,22]. Therefore, *SIX* genes were considered reliable marker genes to distinguish the host specialization of *F. oxysporum.* Thus, combining the results of the pathogenicity experiments, we predict that the *F. oxysporum* saffron strain has a unique combination of *SIX* genes. We searched for homology based on BLAST, and the genome of the *F. oxysporum* saffron strain (XHH35) was predicted to contain seven *SIX* genes (the *SIX 4*, *SIX 6*, *SIX 7*, *SIX 10*, *SIX 11*, *SIX 12*, and *SIX 14* genes) highly homologous to *F. oxysporum* f. sp. *lycopersici*. This prediction was verified by PCR amplification: the correctly sized PCR products were amplified from DNA templates for the *SIX 4*, *SIX 6*, *SIX 7*, *SIX 10*, *SIX 11*, *SIX 12,* and *SIX 14* genes. By constructing the corresponding single-gene evolutionary tree, it was found that in the single-gene evolutionary trees of *SIX 4*, *SIX 6*, *SIX 7*, and *SIX 11*, which did not contain the *F. oxysporum* f. sp. *freesia*, all of the *F. oxysporum* saffron strain (XHH35) genes were on a single branch; in the gene evolutionary trees containing *SIX 10*, *SIX 12*, and *SIX 14* of *F. oxysporum* f. sp. *freesia*, XHH35 was clustered in the same branch as the *F. oxysporum* f. sp. *freesia*. Since the host plant of XHH35-saffron and the host plant of *F. oxysporum* f. sp. *freesia*-freesia are close to each other in terms of species evolutionary relationship, and both belong to the same family of Iridaceae spp., it suggests that the host specialization of *F. oxysporum* may be related to the evolutionary relationship of the host plant, and at the same time, such an association would be reflected in the *SIX* gene. However, the *SIX* genes possessed by the *F. oxysporum* saffron strain (XHH35) and *F. oxysporum* f. sp. *freesia* were different; therefore, the *F. oxysporum* saffron strain in this study was named “*F. oxysporum* f. sp. *Crocus*” in order to reflect its distinct characteristics.

Due to the complexity of *F. oxysporum* complex species, the traditional identification of *F. oxysporum* host specialization is generally complicated by the host crop pathogenicity test. In recent years, with the rapid progress of science and technology, many techniques for rapid detection of plant diseases have emerged. It has been reported that a recombinant recombinase polymerase amplification–lateral flow dipstick (RPA-LFD) has been established for the rapid detection of strawberry wilt [23], which is highly specific for *F. oxysporum*, the causal fungus of strawberry wilt, and can rapidly detect *F. oxysporum* in strawberry wilt samples. In addition, Wang et al. [24] have developed quantitative loop-mediated isothermal amplification (q-LAMP) rapid tests for hickory canker. These tests are of significant value in the detection and prevention of these and other plant diseases. In this study, the specificity of the *F. oxysporum SIX* gene was employed to design eight pairs of primers for PCR amplification, and a pair of primers specific for *F. oxysporum* f. sp. *crocus* was ultimately selected. Following verification of specificity against strains of other species, it was determined that the primers were specific for *F. oxysporum* f. sp. *crocus* strains. This study established a rapid detection and identification method for *F. oxysporum* f. sp. *crocus*, which will be useful in future analyses.

## 4. Materials and Methods

### 4.1. Isolation of Test Strains

This study was performed using the *F. oxysporum* saffron strains isolated and characterized by Ren et al. [4]. In addition, host-specialized strains for tomato, watermelon, and melon were obtained through laboratory isolation and conservation. The strain information is shown in Appendix A.

### 4.2. Pathogenicity Specialization Test

The tomato, melon, and cucumber plants used in the pathogenicity experiment were obtained from seed culture. The seeds of tomato, melon, and cucumber were germinated by moisturizing and germination, and the seedlings were raised on sterilized substrate. After the seedlings had grown several true leaves, the roots were washed with water. Three to five fibrous roots were pressed with sterilized sandpaper, rubbed gently to cause wounds, and then placed into the prepared *F. oxysporum* spore suspension (concentration of 1.0 × 10^6^ units/mL, and the spore suspensions were obtained using the static spore production method [25]) for root immersion treatment for 20 min. The roots were then transferred to the substrate. Immersion in sterile water for 20 min was utilized as a blank control. Each treatment was repeated three times [26]. For saffron and freesia (belonging to the same Iridaceae spp. as saffron), the treatment involved pricking the stems of healthy plants with sterilized insect needles. Prior to inoculation, healthy saffron corms and freesia corms were washed with water and disinfected by wiping the surface with 75% alcohol. Each corm was inoculated with 10 μL of spore suspension (concentration of 1.0 × 10^6^ units/mL). Sterile water was used as a negative control, and three bulbs were inoculated with each strain. Finally, all the above-treated plants were placed in an incubator at 80% humidity and treated at 25 °C under a 12 h light–dark cycle. The plant growth and root development were observed after 7 and 14 days, respectively.

### 4.3. Genome Sequencing and Prediction

After the *F. oxysporum* saffron strain (XHH35) was cultivated on potato dextrose agar (PDA) medium for 10 days, its genomic DNA and messenger RNA were extracted from the mycelium. Genome and transcriptome sequencing were performed using the PacBio and Illumina HiSeq4000 sequencing platforms, respectively, at Biomarker Technologies Co., Ltd. (Beijing, China). Finally, PacBio long reads were obtained for genome assembly, and Illumina short reads were obtained for genome assembly polishing and gene annotation [27].

The query sequence tblastn was used to compare the spliced genome with an E-value of 1 × 10^−5^ and alignment coverage of 35%, after which the macro genome was predicted using getorf X to obtain open reading frames (ORFs) from the spliced genome. The ORFs were compared with the results of the alignment comparison to obtain alignment-complete ORFs.

OrthoVenn3 (https://orthovenn3.bioinfotoolkits.net, accessed on 16 August 2024) was employed to conduct a comparative genomics study on XHH35. Genomic data for three additional host-specialized *Fusarium oxysporum* strains: *Fusarium oxysporum* f. sp. *lycopersici* (GCA_003315725.1), *Fusarium oxysporum* f. sp. *cubense* (GCA_027920445.1) and *Fusarium oxysporum* f. sp. *melonis* (GCA_025216565.1) were obtained from the NCBI. Analysis was conducted based on the OrthoMCL algorithm, and evalue of 1 × 10^−2^ was used [28].

The whole genome sequence data for XHH35 have been deposited in the NCBI (https://www.ncbi.nlm.nih.gov/, accessed on 14 August 2024) under accession number PRJNA1146213.

### 4.4. Amplification Validation and Phylogenetic Analysis of the SIX Gene

The pathogenic fungi were activated on PDA medium, and DNA was extracted from fungal hyphae using the Fungal Genomic DNA Rapid Extraction Kit (Sangon Biotech, Shanghai, China).

*F. oxysporum* 14 *SIX* protein genes. Saffron and freesia are both members of Iridaceae, the iris family. To determine whether the *F. oxysporum* strain isolated from saffron corms was *F. oxysporum* f. sp. *freesia*, the *SIX* genes were used as target genes for phylogenetic analysis. The primers selected for PCR amplification are displayed in Appendix A [20,29]. The PCR reaction system consisted of a total volume of 50 μL, containing 2 μL of each upstream and downstream primer, 1 μL of template, 25 μL of 2×mix TaqDNA Mix, and 20 μL of ddH_2_O. The PCR amplification procedure was as follows: pre-denaturation at 94 °C for 5 min; denaturation at 94 °C for 30 s; annealing based on the temperatures shown in Table 1, with varied annealing times according to the clip length (1000 bp/1 min); and final extension for 10 min at 72 °C. The PCR products were identified on 1.0% agarose gel at 254 nm (ultraviolet) and further sequenced by Shanghai Sangon Biological (Shanghai, China).

The sequencing results were submitted to the NCBI database for comparison and identification to clarify the selection of standard strains. The accession numbers of the standard strains were obtained through a literature search, and their sequence information was downloaded from GenBank [29], as shown in Appendix A. The *SIX* gene sequences were collected from *F. oxysporum* saffron strains that amplified a single target band in PCR, and the *SIX* gene sequences of other strains from different hosts were classified and downloaded. After sequence alignment using MEGA 11, gene evolutionary trees were constructed using the neighbor-joining, maximum likelihood, and maximum resolution methods to map the evolutionary trees of the corresponding *SIX* genes.

### 4.5. Rapid Detection Based on the SIX Gene

Specific primers were designed and screened for the *SIX* gene, which is more specific to *F. oxysporum* saffron strains. PCR amplification of the *SIX* gene was performed using DNA from eight *F. oxysporum* saffron strains and four additional host-specialized strains, and the PCR-amplified products were sent to Shanghai Sangon Biological.

The PCR reaction system had a total volume of 50 μL, consisting of 25 μL of 2×Taq PCR Master Mix, 1 μL each of upstream and downstream primers, 2 μL of template DNA, and 21 μL of ddH_2_O.

The PCR reaction conditions for the *SIX* gene were as follows: pre-denaturation at 94 °C for 5 min, denaturation at 94 °C for 30 s, annealing at 55 °C, extension at 72 °C (with extension times varying according to the lengths of genes with different specificity) for 35 cycles, and final extension at 72 °C for 10 min. Then, 15 μL of PCR amplification products and standard DNA marker were added to 1% agarose gel spotting wells, subjected to electrophoresis at 130 V for 25 min, and then placed in a gel imager to check the electrophoretic bands. The results of specific primer screening were determined according to the agarose gel electrophoresis bands. Finally, strains of different genera were selected to validate the specific primers.

## Figures and Tables

**Figure 1 plants-13-03166-f001:**
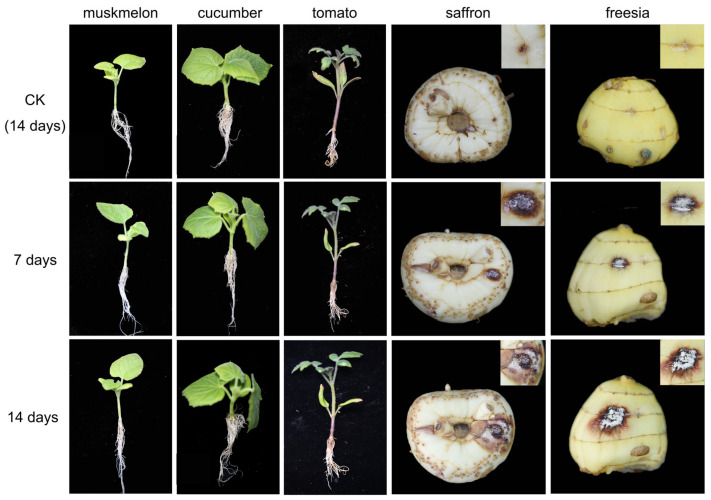
Pathogenicity test of the *Fusarium oxysporum* saffron strain (XHH35) on various plants (muskmelon, cucumber, tomato, saffron, and freesia).

**Figure 2 plants-13-03166-f002:**
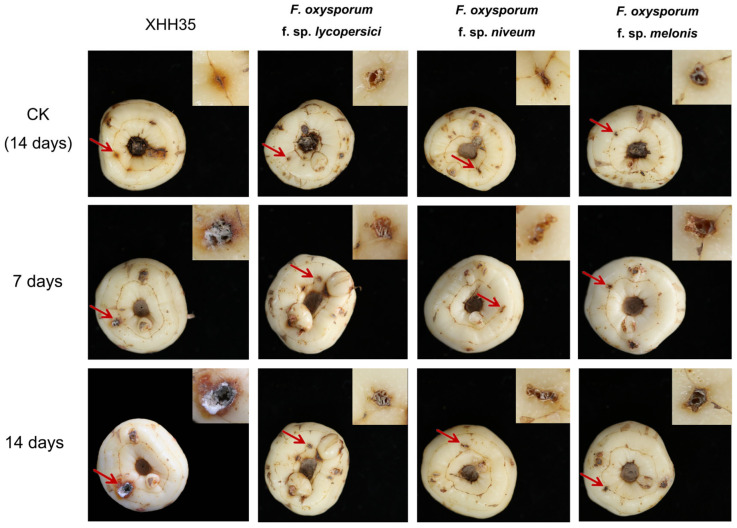
Pathogenicity test of the *Fusarium oxysporum* saffron strain (XHH35) and the other host-specialized strains of *Fusarium oxysporum* (*Fusarium oxysporum* f. sp. *lycopersici*, *Fusarium oxysporum* f. sp. *niveum*, and *Fusarium oxysporum* f. sp. *melonis*) on saffron corms. The red arrow indicates the location where the spore solution was introduced subsequent to a puncture wound with an insect needle.

**Figure 3 plants-13-03166-f003:**
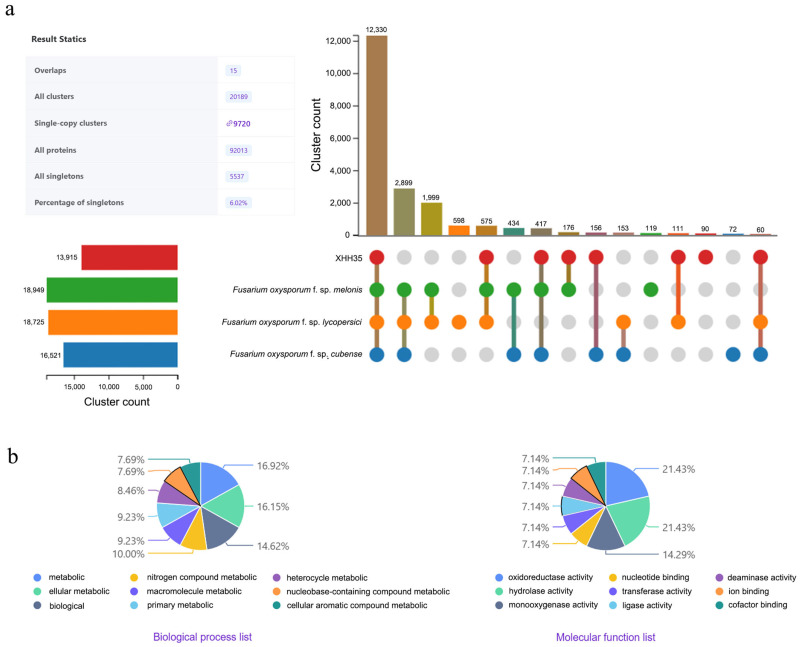
Orthologous cluster analysis of XHH35 and three additional host-specialized strains of *Fusarium oxysporum*: *Fusarium oxysporum* f. sp. *lycopersici*, *Fusarium oxysporum* f. sp. *melonis*, and *Fusarium oxysporum* f. sp. *cubense*. (**a**) UpSet table displaying unique and shared orthologous clusters among the host-specialized strains. The left horizontal bar chart displays the number of orthologous clusters per specialization, while the right vertical bar chart shows the number of orthologous clusters shared among the host specializations. The lines represent intersecting sets; the resulting statics show the total number of sets, clusters, single-copy gene clusters, singletons, and the percentage of singletons across all host specializations of *F. oxysporum*. (**b**) The left and right charts represent the percentage of XHH35-specific genes related to the biological process and molecular function categories, respectively.

**Figure 4 plants-13-03166-f004:**
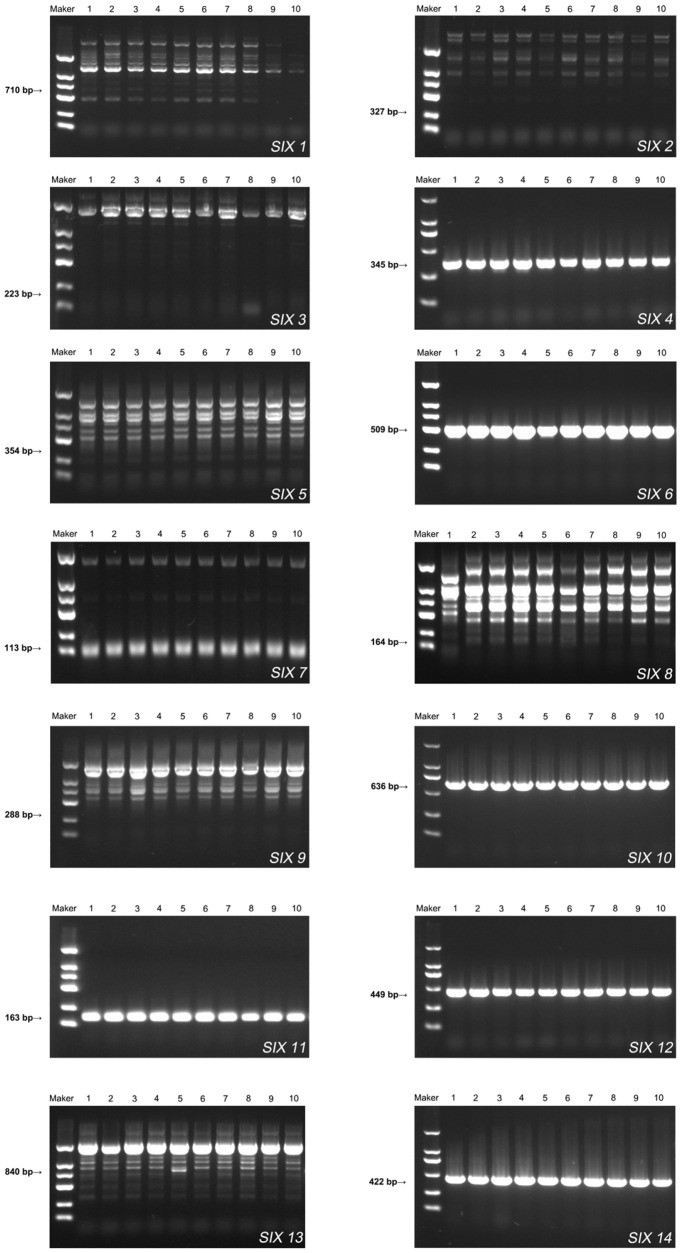
Gel electrophoresis results of the *SIX 1–14* gene amplification products of *Fusarium oxysporum* saffron strains: 1–10: *F. oxysporum* saffron strains (1: XHH2; 2: XHH16; 3: XHH35; 4: XHH47; 5: XHH56; 6: XHH66; 7: XHH67; 8: XHH76; 9: XHH78; 10: XHH80) [4]; M: DNA Molecular Weight Standard Marker (100~2000 bp; Sangon Biotech, Shanghai, China).

**Figure 5 plants-13-03166-f005:**
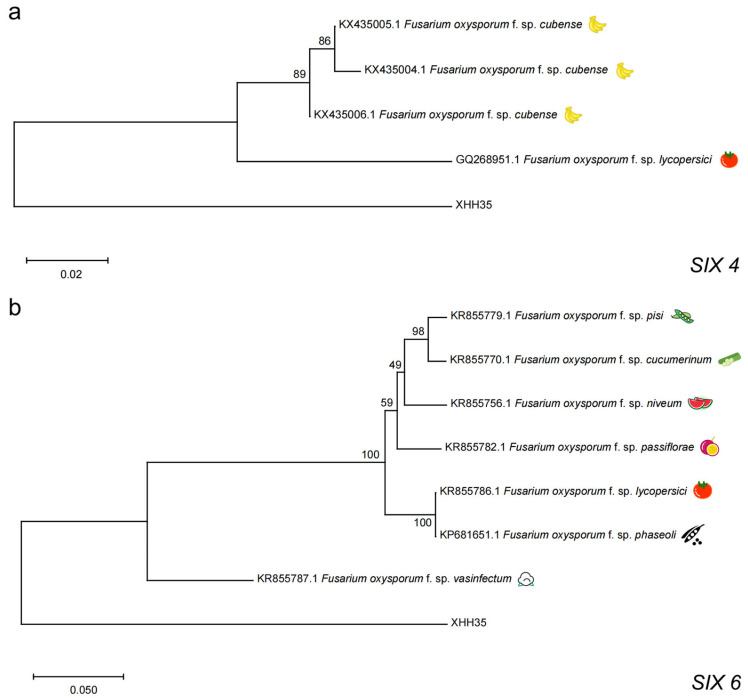
Phylogenetic tree of the *Fusarium oxysporum* saffron strain (XHH35) with the *SIX 4* (**a**), *SIX 6* (**b**), *SIX 7* (**c**), *SIX 10* (**d**), *SIX 11* (**e**), *SIX12* (**f**) and *SIX14* (**g**) genes of other host-specialized strains.

**Figure 6 plants-13-03166-f006:**
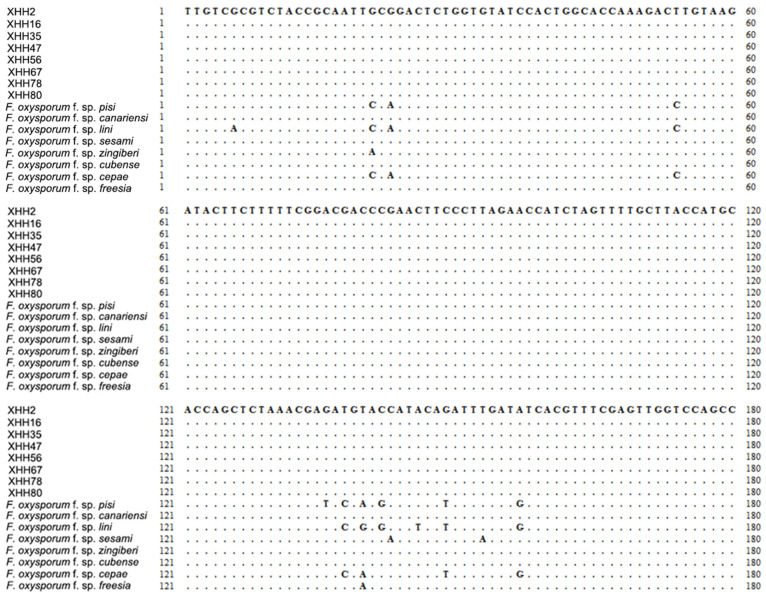
Sequence comparison of *SIX 10* genes (partial sequences) of different formae speciales.

**Figure 7 plants-13-03166-f007:**
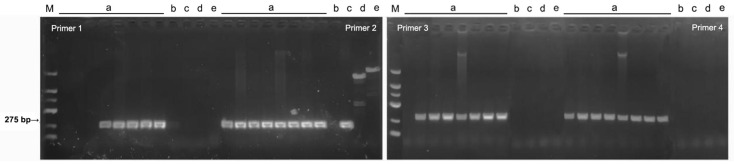
Gel electrophoresis with 1% agarose of candidate primers for host-specialized strains of *Fusarium oxysporum* a: *F. oxysporum* saffron strains (1: XHH2; 2: XHH16; 3: XHH35; 4: XHH47; 5: XHH56; 6: XHH66; 7: XHH67; 8: XHH76); b: *F. oxysporum* f. sp. *fragariae*; c: *F. oxysporum* f. sp. *lycopersici*; d: *F. oxysporum* f. sp. *niveum*; e: *F. oxysporum* f. sp. *melonis*; M: DNA Molecular Weight Standard Marker (100~2000 bp; Sangon Biotech, Shanghai, China).

**Figure 8 plants-13-03166-f008:**
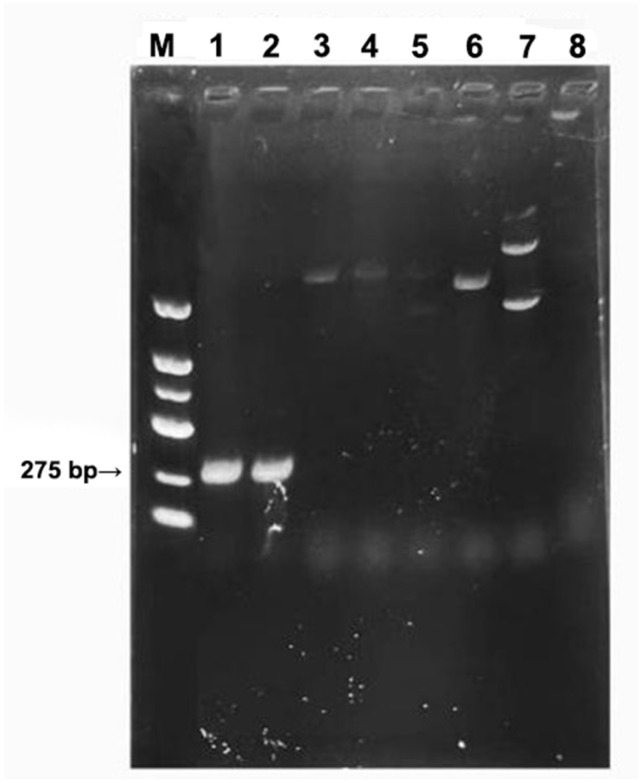
Gel electrophoresis map of specific detection of other pathogens using primer 4: 1–2: *F. oxysporum* saffron strains (1: XHH16; 2: XHH35); 3: *Fusarium fujikuroi*; 4: *Colletotrichum*; 5: *Alternaria alternata*; 6: *Botryosphaeria dothidea*; 7: *Corynespora*; 8: *Nigrospora*; M: DNA Molecular Weight Standard Marker (100~2000 bp; Sangon Biotech, Shanghai, China).

**Table 1 plants-13-03166-t001:** Genome features of the *Fusarium oxysporum* saffron strains (XHH35).

Genome Assembly Info	Count
Sequencing technology	PacBio + Illumina
Scaffold number	34
N50 (bp)	4,241,186
N90 (bp)	2,232,100
GC content (%)	48
Genome size (bp)	56,897,855
Protein-coding genes	15,461
Transcript number	15,684
exon number	43,958

**Table 2 plants-13-03166-t002:** GO Enrichment result (*p*-value of 1 × 10^−2^).

GO Term Category	Name	Count	*p*-Value
Biological process	Transmembrane transport	7	1.01 × 10^−63^
Biological process	Transcription (DNA-templated)	7	1.79 × 10^−50^
Biological process	Sporulation resulting in formation of a cellular spore	2	2.41 × 10^−17^
Molecular function	Oxidoreductase activity (acting on paired donors, with incorporation or reduction of molecular oxygen)	6	4.29 × 10^−7^
Biological process	Regulation of transcription (DNA-templated)	3	1.45 × 10^−5^
Biological process	Carbohydrate transport	2	1.94 × 10^−5^
Biological process	Positive regulation of transcription from RNA polymerase II promoter	2	1.02 × 10^−4^
Biological process	Polyketide biosynthetic process	5	1.82 × 10^−4^
Biological process	Carbohydrate metabolic process	2	2.76 × 10^−4^

**Table 3 plants-13-03166-t003:** Specific candidate primers for *Fusarium oxysporum* of saffron based on *SIX 10*. In the software Primer Premier v5.0, which is used to design primers, the rating indicates the degree of fit of the primers. The primers that were determined to be the most suitable based on the software’s design criteria are indicated in the table by the designation “Primers 1–8”.

Number	Primers (5’-3’)		Rating (%)	Product Size (bp)	Annealing Temperature (°C)
1	Primer 1-F	ATTGCGGACTCTGGTGTA	100	275	51.5
	Primer 1-R	TGAAAGCGTTGTAATGTTG			
2	Primer 2-F	TGCGGACTCTGGTGTATC	100	277	51.8
	Primer 2-R	CCTCTGAAAGCGTTGTAAT			
3	Primer 3-F	TGCGGACTCTGGTGTATC	98	273	51.6
	Primer 3-R	TGAAAGCGTTGTAATGTTG			
4	Primer 4-F	TGCGGACTCTGGTGTATC	96	275	51.2
	Primer 4-R	TCTGAAAGCGTTGTAATGT			
5	Primer 5-F	ATCCACTGGCACCAAAGA	96	334	52.5
	Primer 5-R	TCAGGGTAGACACCAAATCG			
6	Primer 6-F	CCACTGGCACCAAAGACT	96	332	52.4
	Primer 6-R	TCAGGGTAGACACCAAATCG			
7	Primer 7-F	GACCCGAACTTCCCTTAG	96	258	51.3
	Primer 7-R	TACTGGTTGTAGCCGTGA			
8	Primer 8-F	ACGACCCGAACTTCCCTT	86	250	50.4
	Primer 8-R	AGCCGTGAGTAAATGTGA			

## Data Availability

Whole genome sequence data for XHH35 have been deposited in the NCBI (https://www.ncbi.nlm.nih.gov/) under accession numbers PRJNA1146213.

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
