# Peer review of "Characterization, Genome Sequencing, and Development of a Rapid PCR Identification Primer for Fusarium oxysporum f. sp. crocus, a New forma specialis Causing Saffron Corm Rot"

_plants, 2024, doi:10.3390/plants13223166_

Round 1
Reviewer 1 Report
Comments and Suggestions for Authors
Dear Authors
General the manuscript is well-structured and the information presented in a logical manner but some information in sections Materials and Methods and Results and Discussion must be completed.
I have made 13 queries/comments throughout the manuscript. Therefore, the present draft needs revision before further process.

Author Response
General the manuscript is well-structured and the information presented in a logical manner but some information in sections Materials and Methods and Results and Discussion must be completed.
I have made 13 queries/comments throughout the manuscript. Therefore, the present draft needs revision before further process.
Response: Thanks for the suggestion. I'm responding to your 13 queries/comments:
- Line 77: oxysporum saffron strain: XHH35 (Strain number). I have added this information to the text.
- Line 95: oxysporum saffron strain (XHH35). I have added this information to the text.
- Line 100: oxysporum saffron strain: XHH35 (Strain number). I have added this information to the line 77.
- Line 140: 1–10: oxysporum saffron strains (1: XHH2; 2: XHH16; 3: XHH35; 4: XHH47; 5: XHH56; 6: XHH66; 7: XHH67; 8: XHH76; 9: XHH78; 10: XHH80). I have added this information to the annotate.
- Line 141: DNA Molecular Weight Standard Marker (100~2000 bp; Sangon Biotech, Shanghai, China). I have added this information to the annotate.
- Line 221: 1–10: oxysporum saffron strains (1: XHH2; 2: XHH16; 3: XHH35; 4: XHH47; 5: XHH56; 6: XHH66; 7: XHH67; 8: XHH76). I have added this information to the annotate.
- Line 222: DNA Molecular Weight Standard Marker (100~2000 bp; Sangon Biotech, Shanghai, China). I have added this information to the annotate.
- Line 230: 1-2: oxysporum saffron strains (F. oxysporum f. sp. crocus; 1: XHH16; 2: XHH35). I have added this information to the annotate.
- Line 232: Thank you for your suggestion. I have reconsidered the discussion section of the article in more detail in the context of the literature.
- Line 265:1-2 is oxysporum f. sp. crocus (1: XHH16; 2: XHH35).
- Line 279: oxysporum spore suspension (Concentration of 1.0 x 106 units/mL). I have added this information to the text.
- Line 283: Belonging to the same Iridaceae spp. as saffron. I have added this information to the text.
Line 286: Spore suspension (Concentration of 1.0 x 106 units/mL). Also, with only one puncture treatment on a saffron corm, the saffron corm eventually develops only at the puncture treatment. I have marked the puncture treatment sites in the experiment with red arrows (Fig. 2).
Reviewer 2 Report
Comments and Suggestions for Authors
Dear authors,
This paper entails a very interesting study on the identification and confirmation and pathogenicity of the host specialisation forms of the soil-borne pathogen, Fusarium oxysporum, that causes corm rotting in saffron. PCR diagnostics and genomic resources were developed in this study.
I believe the experiments performed in this study are scientifically sound. But the presentation of the results needs a major improvement and so does the discussion section of this manuscript.
To help the authors revise the manuscript, I've listed point-by-point, the sections that need improvements below.
Line 13: Please rephrase this sentence so that it does not start with 'Previous studies'.
Line 23: Italicise 'formae speciales'.
Line 29: what is the meaning of 'self-contained' here? Please rephrase this sentence.
Line 30: 'the first to report' should be 'the first ever report of'
Line 37: 'affects' should be 'improves'.
Line 45: 'The typical symptoms of SCR in the fields are wilted yellow leaves' should be 'The typical symptoms of SCR in the field include leaf yellowing and wilting.
Line 54: 'For example, watermelon and melon, both members of the cucurbit family, have distinct host specialization' - Please rephrase this - plant hosts cannot have distinct host specializations.
Line79-8: I have a small issue with the pathogenicity testing here. With the melon, cucumber, and tomato inoculated plants, you examined the leaf growth (wilting) whereas for both saffron and freesia, you examined SCR in the corms - the comparison of symptoms needs to be consistent across different plant species. I suggest that you firstly describe wilting of leaves for all plant species tested, then go on to describe the SCR in the corms.
line 90: Change 'control' to 'non-inoculated' group.
Line 94: Figure 1. please add symptomatic plants of saffron and freesia to make side-by-side comparisons with muskmelon, cucumber and tomato.
Line 98. Figure 2. Please add arrows to the presence of the fungus in the symptomatic regions of saffron corms and explain this in the figure legend.
Line 105: You mentioned 'PacBio long reads' then in Table 1 (line 109) you stated that the sequencing technology was 'ONT + Illumina'. Did you use PacBio or ONT? Please clarify.
Line 110 - 'A comparative study' - exactly what was done here? please provide sufficient information to describe this result.
Line 113 - 'other three host-specialized strains of F. oxysporum' - please specify what they are here.
Line 113-116 - More details are needed to describe the functional categorisations - is it based on the GO terms? You've not mentioned the analysis of Gene Ontology enrichment here and the information relating to the GO terms described in Figure 3B.
Line 132 - 'Polymerase chain reaction (PCR) amplification was performed based on the genomic prediction results.' should be '... to further confirm the presence of these genes in XHH35.'
Line 135: 'F. oxysporum saffron could amplify..' should be ' PCR products of the correct size were amplified from the DNA template of xxx for ...'
Line 140: In the figure legend and in the corresponding results, please explain the origin of the 10 strains. Are they different to XHHH35?
Lin 152-154. The wording is just not clear on what was done. Please rephrase.
Figure 5 (Line 195-197). There is a major issue with the phylogenetic trees. The size of the text on all trees presented is too small to be legible. All trees should be re-done with an acceptable font size including the scale bars too.
Figure 6 (line 203-204). Same issue as Figure 5. Font and sizes should be standardised here.
Line 205 (Figure 6 legend): Figure legend does not provide sufficient information on the alignment program. The sequence of SIX 10 genes - does the MSA span the entire gene? Please indicate the positions of the alignment with respect to the start (ATG) and the stop codon of the SIX10 gene.
Table 2. What does 'rating mean'. Please describe this and the rest of what's presented here in the table legend. Primer pairs 1-8, I would like a diagrammatic scheme to show which regions of the SIX10 genes were amplified with these primers.
Figure 7 (line 208-209): The images are blurry. Was there a problem with the running buffer? Especially for the bottom gels, the PCR products are showing 'doubles' so it is not clear whether a single product was obtained. I recommend moving the bottom two gel images to the supplementals.
For the Figure 7 legends, 'a b c d e' categories should be annotated on the bottom gels too. What DNA ladder (size and brand) was used? Is this an agarose gel?
Figure 8. This figure was not cited in the results or discussion. Please cite this figure in the text and discuss the relevant points accordingly.
What is the DNA marker used? and please explain the other PCR products obtained from lanes 6, 7, and 8. What are these products? are these non-specific products or did the primers amplified an orthologous region in these other fungal species.
Discussion 233-266: The current discussion is not a discussion. It's just a re-count of the results. There are very few references cited in the discussion. It is just not comprehensive enough to clarify the relevance of the findings from this study to this particular field of research. Therefore, discussion needs to be re-written based on the pathogenicity of FOs, the usefulness of this assembly as compared to the other assemblies available in the public domain, and FO diagnostics, and how this system can aid the detection of the pathogen - all of this should be sufficiently discussed and backed up with current and historical references.
Line 305-307: please list the genome accessions for these assemblies.
Author Response
This paper entails a very interesting study on the identification and confirmation and pathogenicity of the host specialisation forms of the soil-borne pathogen, Fusarium oxysporum, that causes corm rotting in saffron. PCR diagnostics and genomic resources were developed in this study.
I believe the experiments performed in this study are scientifically sound. But the presentation of the results needs a major improvement and so does the discussion section of this manuscript.
To help the authors revise the manuscript, I've listed point-by-point, the sections that need improvements below.
Line 13: Please rephrase this sentence so that it does not start with 'Previous studies'.
Response: Thanks for the suggestion. This sentence was revised as: 'Saffron corm rot (SCR), the most serious disease affecting saffron, has been confirmed to be caused by Fusarium oxysporum in previous studies.'
Line 23: Italicise 'formae speciales'.
Response: Thanks for the suggestion, it's been revised.
Line 29: what is the meaning of 'self-contained' here? Please rephrase this sentence.
Response: Thanks for the suggestion. This sentence was revised as: 'The corresponding individual phylogenetic tree indicated that F. oxysporum saffron strain (XHH35) showed a separate branch with different formae speciales.'
Line 30: 'the first to report' should be 'the first ever report of'
Response: Thanks for the suggestion, it's been revised.
Line 37: 'affects' should be 'improves'.
Response: Thanks for the suggestion, it's been revised.
Line 45: 'The typical symptoms of SCR in the fields are wilted yellow leaves' should be 'The typical symptoms of SCR in the field include leaf yellowing and wilting.
Response: Thanks for the suggestion. This sentence was revised as: 'The typical symptoms of SCR in the field include leaf yellowing and wilting on the ground, as well as atrophied rotting corms underground.'
Line 54: 'For example, watermelon and melon, both members of the cucurbit family, have distinct host specialization' - Please rephrase this - plant hosts cannot have distinct host specializations.
Response: Thanks for the suggestion, it's been revised.
Round 2
Reviewer 2 Report
Comments and Suggestions for Authors
Dear authors, your manuscript has been overhauled and it is now much improved. With a few outstanding (minor) points below requiring your attention, I find this manuscript acceptable for publication.
1) line 54-55: please delete 'Plant hosts cannot have distinct host specializations', and add 'For example, the pathogen forms of F. oxysporum have distinct host specialisation on water-melon and melon, both of which are members of the cucurbit family [9]'.
2) line 54-55: reference number 9 or [9] appears to be in Chinese. Therefore, I encourage the authors to find other references (in English) to replace this reference.
3) Table 2 (Line 133). Please change the heading 'Name space' to 'GO term category'.
4) Line 258-260. Please move Figure 7 (primer positions on the SIX10 gene sequence) into the supplemental.
Author Response
Dear authors, your manuscript has been overhauled and it is now much improved. With a few outstanding (minor) points below requiring your attention, I find this manuscript acceptable for publication.
- line 54-55: please delete 'Plant hosts cannot have distinct host specializations', and add 'For example, the pathogen forms of oxysporum have distinct host specialisation on water-melon and melon, both of which are members of the cucurbit family [9]'.
Response: Thanks for the suggestion, it's been revised.
- line 54-55: reference number 9 or [9] appears to be in Chinese. Therefore, I encourage the authors to find other references (in English) to replace this reference.
Response: Thanks for the suggestion, I have replaced this reference (in Chinese) with a new one (in English).
- Table 2 (Line 133). Please change the heading 'Name space' to 'GO term category'.
Response: Thanks for the suggestion, it's been revised.
4) Line 258-260. Please move Figure 7 (primer positions on the SIX10 gene sequence) into the supplemental.
Response: Thanks for the suggestion, I have moved Figure 7 into the supplemental.